# Qualitative Analysis of Visible Foreign Solids in Armillarisin A Injection Formulations Using Ultra-High Performance Liquid Chromatography–Tandem Mass Spectrometry

**DOI:** 10.3390/molecules28041609

**Published:** 2023-02-07

**Authors:** Ruiqi Wang, Haichang Zhou, Shiyu Liao, Qi Tian, Zhengbing Lv, Kangde Bao, Lili Liu

**Affiliations:** 1College of Life Sciences and Medicine, Zhejiang Sci-Tech University, Hangzhou 310018, China; 2Zhejiang Provincial Key Laboratory of Silkworm Bioreactor and Biomedicine, Zhejiang Sci-Tech University, Hangzhou 310018, China; 3Technical Centre, Jiangxi Qingfeng Pharmaceutical Co., Ltd., Ganzhou 341008, China

**Keywords:** Armillarisin A injection, visible foreign matter, ultra-high performance liquid chromatography–tandem mass spectrometry, X-ray powder diffraction

## Abstract

During the trial production of Armillarisin A for injection (AA-I), unidentified needle-like yellow-brown crystals were occasionally observed. Here, we report an ultra-high performance liquid chromatography–tandem mass spectrometry (UPLC-MS) method for determining the source of the visible foreign bodies in the formulations of Armillarisin A active pharmaceutical ingredient (AA-API). AA-API, photolyzed samples, the intermediate polymer, and the excipient analyzed determined after the separation on a Waters Symmetry C18 (3.5 μm, 4.6 × 75 mm) column with a mobile phase consisting of a methanol/acetic acid (0.1 mol/L) aqueous solution (50:50). Furthermore, the crystal type of the visible foreign bodies, the intermediate polymer and AA-API were investigated by X-ray powder diffraction (XRD). The results revealed that the characteristics of the visible foreign solids were the same as those of AA-API as regards UPLC peak position (368 nm) and MS spectrum in negative ion detection mode. The visible foreign solids were thus identified as unpolymerized crystals of AA-API and were attributed to AA-API itself. The results showed that the production process could be improved by changing the stirring method and frequency as well as by optimizing the polymerization temperature to ensure the safety, stability, and control of the product quality in the stage of batch production.

## 1. Introduction

Armillarisin A (molecular formula C_12_H_10_O_5_, molecular weight 234.21) is a new coumarin (3-acetyl-5-hydroxylmetyl-7-hydroxycoumarin) prepared from the mycelium of *Armillariella tabescens* [1]. It consists of yellow or slightly orange rectangular plate-like crystals or a crystalline powder that is odorless, insoluble in water, and slightly soluble in ethanol or methanol [2]. The Armillarisin A injection (AA-I) formulation is produced by a modern process and is used clinically for its choleretic action, high bioavailability, unique mechanism, precise clinical efficacy, and limited side effects. It is applied for treating acute cholecystitis, chronic cholecystitis, some biliary tract diseases caused by acute infection and chronic superficial gastritis, and chronic superficial atrophic gastritis [3,4,5]. A research report published by the editorial board of the Chinese Journal of Gastroenterology and the Collaborative Group of Hepatobiliary Diseases of the Chinese Society of Gastroenterology revealed that with the gradual improvement of the living standard of people in China, the incidence of chronic cholecystitis and gallstones has been increasing in recent years [6]. As a result, AA-I is expected to be more largely used in future clinical applications for its therapeutic effects.

Generally, with promulgated drug quality standards, drug manufacturers can only formulate standard operating procedures after optimizing and verifying the production process parameters within the framework of the legal standards, which can be implemented after being approved by the national drug administration department. According to the Chinese national pharmaceutical standard [No. WS-10001-(HD-1207)-2002] of AA-I, the solvent for dissolving Armillarisin A can be selected by each manufacturer. Most commercially available AA-I contain pharmaceutical-grade propylene glycol as a solvent. When used in high doses, it sometimes causes propylene glycol toxicity [7]. Lei Li et al. [8] encapsulated Armillarisin A into lipid nanoparticles to ensure that the drug particles dissolved and existed under molecular form in the lipid-like material, which improved its solubility and stability under the premise of ensuring safety. However, since Armillarisin A is poorly soluble in water, it is imperative to find a suitable cosolvent for AA-I production. Povidone K30, a white-to-creamy powder, is a slightly odorous, tasteless organic compound with the molecular formula (C_6_H_9_NO)_n_ and is often used as a polymerizing agent what can embed Armillarisin A molecules into its molecular framework in the production of AA-I to promote the dissolution of Armillarisin A. This leads to advantages, such as higher solubility and stability, with respect to other available formulations [9], but occasionally, needle-like yellow-brown crystals were observed in some of the AA-I trial production samples [2,5]. There is no doubt that visible foreign bodies are absolutely prohibited for injections, because they directly endanger the safety of the patients. If the visible foreign bodies cannot be eliminated, this will not only seriously affect the product quality and safety, but also lead to the failure of the subsequent batch production. It is a great necessity to trace the source of these bodies and confirm their identity, searching their possible causes and optimizing and verifying the batch production process. The current study aimed to qualitatively identify the visible foreign solids by comparing the UPLC-MS chromatograms and spectra of the Armillarisin A active pharmaceutical ingredient (AA-API), photolyzed samples, the intermediate polymer, and the excipient, as well as by comparing the X-ray powder diffraction (XRD) spectra of the intermediate polymer and AA-API. Only when the source and identity of the visible foreign solids are determined in the trial production stage, can effective measures be taken to ensure the safety, stability, and control of the product quality in the stage of AA-I batch production.

## 2. Results

The UPLC chromatograms (368 nm) of AA-API, a photolyzed sample, visible foreign solids, intermediate polymer, and blank excipient are shown in Figure 1.

For both AA-API and the visible foreign solids, secondary scans of the quasi-molecular ion [M-H]^-^ at *m*/*z* 233 were further performed, and the ESI-MS^2^ scan mass spectra (Figure 2) revealed that the [M-H]^-^ parent ion mainly yielded 10 fragment ions with relatively large abundances, including ions with *m*/*z* of 214.99, 203.06, 191.02, 189.02, 171.02, 162.99, 161.01, 158.95, 147.01, and 135.01.

The presumed cleavage pathway of AA-API in negative ion detection mode, together with the chemical structure of AA-API in negative ion mode as [M-H]^-^, are shown in Figure 3.

## 3. Discussion

### 3.1. Investigating the Source of the Visible Foreign Bodies

Since the pharmaceutical registration process has global validity, the actual manufacturing process first needs to be converted into an operational standard operating procedure (SOP). The solvent used in the preparation of standard AA-I can vary. However, the detailed formulation and process parameters are commercial secrets belonging to the producing enterprises. They are generally not shared with the public but only with the national regulatory authorities.

According to the formulation composition and production process recorded in the Chinese national pharmaceutical standard [No. WS-10001-(HD-1207)-2002] of AA-I, a trial production was carried out. Pharmaceutical-grade povidone K30 was chosen as a cosolvent, as it provided more advantages than those obtained with different cosolvents by other manufacturers, such as high solubility and stability. The samples used in this study were all from the trial production batch (5000 ampoules, 10 mL each); those containing visible foreign bodies were collected for further investigation.

Based on the trial production process, AA-API, the polymerization agent (povidone K30), the solvent (anhydrous ethanol, N,N-dimethylacetamide, propylene glycol, polyethylene glycol 400, polyethylene glycol 6000), the surfactant (polysorbate 80), and the antioxidant (thiourea) were employed. Among these compounds, only Armillarisin A and povidone K30 are crystalline. Owing to the formulation composition, the production process, and the type and structural characteristics of the raw and auxiliary materials used in the Chinese national pharmaceutical standard [No. WS-10001-(HD-1207)-2002] of AA-I, the visible foreign solids may derive from either API (Armillarisin A) or the polymerization agent (povidone K30), but not from the solvent (anhydrous ethanol, N,N-dimethylacetamide, propylene glycol, polyethylene glycol 400, polyethylene glycol 6000), the surfactant (polysorbate 80), or the antioxidant (thiourea).

Furthermore, AA-I can be decomposed by various factors such as light, high temperature, and pH [10,11]. The UV characteristic spectra of the photolyzed samples, the unidentified solid, and the intermediate polymer at 368 nm were highly consistent with that of AA-API, while the blank excipient did not show any significant absorption at 368 nm (as shown in Figure 1). The possible source of the visible solids could be tentatively inferred by combining these results with those already presented in the literature [12,13,14,15].

### 3.2. Detection and Analysis of AA-API and the Unidentified Solid by UPLC-MS/MS

Based on the above tentative analysis of the source of the visible foreign bodies, AA-API and the visible foreign bodies were detected by LC-MS/MS and analyzed spectroscopically. The ESI-MS spectra of AA-API revealed that AA-API was prone to lose 1 H^+^ in negative ion detection mode, thus forming the quasi-molecular ion [M-H]^-^ at *m*/*z* 233.08 (Figure 3) in large abundance. It indicated that the AA-API quasi-molecular ion ESI-MS existed as [M-H]^-^ and had better ionic stability in the negative ion mode.

Figure 3 shows that the quasi-molecular ion [M-H]^-^ at *m*/*z* 233 lost a molecule of HCHO, yielding a fragment ion at *m*/*z* 203.06 (M1). M1 was cleaved in turn, losing a molecule of CH_2_=C=O, and CO yielded the fragment ions at *m*/*z* 161.01 and *m*/*z* 135.01. The quasi-molecular ion [M-H]^-^ at *m*/*z* 233 lost a molecule of CO_2_, yielding a fragment ion t *m*/*z* 189.02 (M2). M2 could lose a molecule of H_2_O, yielding a fragment ion with an *m*/*z* of 171.02, a molecule of CH_2_=C=O, yielding a fragment ion at *m*/*z* 147.01, or a molecule of HCHO, yielding a fragment ion at *m*/*z* 158.95. The quasi-molecular ion [M-H]^-^ at *m*/*z* 233 lost one molecule of H_2_O to form a fragment ion at *m*/*z* 214.99 (M3); the quasi-molecular ion [M-H]^-^ at *m*/*z* 233 lost one molecule of CH_2_=C=O to form a fragment ion at *m*/*z* 191.02 (M4). M4 lost one molecule of H_2_O, producing a fragment ion at *m*/*z* 162.99, and the further loss of one molecule of H_2_O produced a fragment ion at *m*/*z* 147.01.

The peak position of the visible foreign solids was consistent with that of AA-API, according to the respective UPLC chromatograms and MS spectra (Figure 4). The visible foreign solids were identified as the crystallized product of AA-API. Based on the secondary mass spectrometry results of UPLC-MS/MS for the AA-I product and the visible unidentified solids, combined with the analysis of possible cleavage pattern of AA-API [16,17], it was presumed that the occasional visible foreign bodies were precipitated API or not fully polymerized and wrapped API.

### 3.3. Analysis of the Causes of the Visible Foreign Bodies

XRD analysis was performed in the 3–90° range [18]. The results are shown in Figure 5. AA-API was copolymerized with povidone K30 at a 1:5 ratio at 50 °C, which resulted in the loss of its original crystallinity and the acquisition of an amorphous form in the povidone skeleton. The recrystallization of API was mainly inhibited by the intermolecular interaction force between the two materials.

The crystal type of the visible foreign bodies was very different from that of the intermediate polymer and of AA-API (Figure 5). The position and intensity of the diffraction peaks were significantly different. Combined with the results of the structure identification of the visible foreign solids, it was initially inferred that the formation of the visible foreign solids might be caused by some specific factors (such as trace impurities, a critical temperature, etc.) and triggered by the recovery of a certain type of crystalline components presented in API and derived from the copolymer and the precipitation process (shown in the oval circle in Figure 5).

## 4. Materials and Methods

### 4.1. Reagents and Excipients

Methanol (HPLC grade) was purchased from Scharlau (Barcelona, Spain), acetic acid (HPLC grade) was purchased from Aladdin Holdings Group Co. (Shanghai, China) and ammonia was obtained from Zhejiang Hanno Chemical Technology Co. Ltd. (Lanxi, China). Purified water was prepared by a Milli-Q water purification system (Merck, Darmstadt, Germany). The reagents and excipients used in AA-I production, such as povidone K30, were of medicinal grade and met the requirements of the Chinese Pharmacopoeia or the national drug quality standards.

### 4.2. Preparation of the Solutions

#### 4.2.1. AA-API, Photolytic Samples, and Intermediate Polymer Solutions

The photolytic samples were prepared by spreading AA-API (2 g, purchased from Shanxi Tongji Pharmaceutical Co., Ltd., Yuncheng, China) in a dish and then placing it in a lightbox (6000 Lux) for 24 h; it was finally stored in a closed and shaded environment. The intermediate polymer was prepared by mixing AA-API and povidone K30 in appropriate amounts, and the intermediate polymerization was performed according to the Chinese national drug standard of AA-I. AA-API (12.5 mg), the photolyzed samples (12.5 mg), and the intermediate polymer (12.5 mg) were separately placed in volumetric flasks (50 mL), with methanol added to adjust the volume. A sample with a concentration of 5 μg/mL was obtained by pipetting 1 mL of the solution into a 50 mL volumetric flask and adjusting the volume with methanol.

The blank excipient was prepared from anhydrous ethanol, N, N-dimethylacetamide, propylene glycol, polyethylene glycol 400, polyethylene glycol 6000, polysorbate 80, and thiourea, according to the preparation method of AA-I recorded in the Chinese national drug standards [19]. The blank excipient (1 mL) was pipetted into a volumetric flask (10 mL), with the volume adjusted using methanol.

#### 4.2.2. Preparation of the Ammonia Test Solution and the Foreign Body Sample Solution

As reported in the Chinese national drug standard of Armillarisin A injection, an ammonia solution was used for a color reaction. Thus, the ammonia solution was used to try to dissolve the visible foreign bodies which were insoluble in water. The ammonia test solution was prepared by adding concentrated ammonia (400 mL) to a volumetric flask (1000 mL), and the volume was adjusted by adding water. Due to the stability test results that showed that two degraded components could be observed 2 h later under the same UPLC conditions, the ammonia solution was prepared only before use in this study as a cosolvent.

Some AA-I formulations were selected from the trial production and left standing for 2 min. A syringe was then used to aspirate the upper clear layer, while the large needle-like crystals at the bottom were retained. The blank auxiliary solution (0.5 mL) was aspirated by another syringe to rinse the inner wall of the ampoule, the ampoule was shaken slowly and left standing still until the upper layer solution was clear and could be aspirated. This process was repeated twice, and a change in the crystal properties during this process was observed. Methanol (0.5 mL) was then aspirated to rinse the inner wall and the needle crystals in the same way. This process was repeated twice, and the solution was left standing while the upper clear layer was aspirated. An ammonia/water (0.5 mL, 1:4) solution was added to the residual crystals, the mixture was shaken until the crystals disappeared, and then it was diluted with methanol to obtain the unidentified visible crystal sample solution (1 mL).

### 4.3. Sample Weighing

The samples were accurately weighed using a QUINTIX35-1CN 1-in-100,000 electronic balance (Sartorius, Goettingen, Germany).

### 4.4. UPLC-MS/MS Conditions

Ultra-high performance liquid chromatography analysis was performed on an Acquity I-Class PLUS ultra-high performance liquid chromatographer (Waters) with an Acquity/Xevo TQS ultra-high performance liquid chromatographer–mass spectrometer (Waters) coupled with an electrospray ionization source, in negative ion detection mode. The source temperature was 150 °C, the capillary voltage was 2 kV, the cone voltage was 30 V, and the cone hole gas flow rate was 150 L/hr. The shell gas temperature was 350 °C and the shell gas flow rate was 650 L/hr. The carrier gas flowed at 0.7 MPa at a collision energy of 25 V. The separation was performed on a Waters Symmetry C18 column (3.5 μm, 4.6 × 75 mm) with a methanol −0.1 mol/L acetic acid solution (50:50) at a flow rate of 0.4 mL/min and a column temperature of 25 °C.

### 4.5. XRD Conditions

The XRD analysis was performed on a RIKEN X-ray multi-crystal diffractometer (XRD Rigaku Smart-Lab 9 kW) at an operating capacity of 1.6 kW (40 kV, 40 mA) within the scanning range of 3~90° at a step distance of 0.02. The target material was Cu.

## 5. Conclusions

In conclusion, this study enabled the qualitative identification of the visible foreign bodies, occasional needle crystal-like yellow-brown precipitates, observed in the trial production of AA-I samples, by comparing the UPLC-MS chromatograms and spectra of the visible foreign solids with those of AA-API, photolyzed samples, the intermediate polymer, and the excipient, as well as by combining the LC-MS/MS and XRD information. The results revealed that the visible foreign solids consisted of a small amount of AA-API ingredients not encapsulated by the povidone K30 copolymer. These results can be beneficial for improving the production technology and the quality control in drug production enterprises and can help quickly solve similar problems that may occur in production processes. To sum up, optimizing the preparation temperature, changing the stirring method and the stirring frequency, and conducting a verification process can further improve a product quality and ensure its safety for clinical use [20].

## 6. Patents

There is a patent named “A method for identifying endogenous visible foreign bodies in Armillarisin A injection”, with registration No. 202211654876.9, resulting from the work reported in this manuscript.

## Figures and Tables

**Figure 1 molecules-28-01609-f001:**
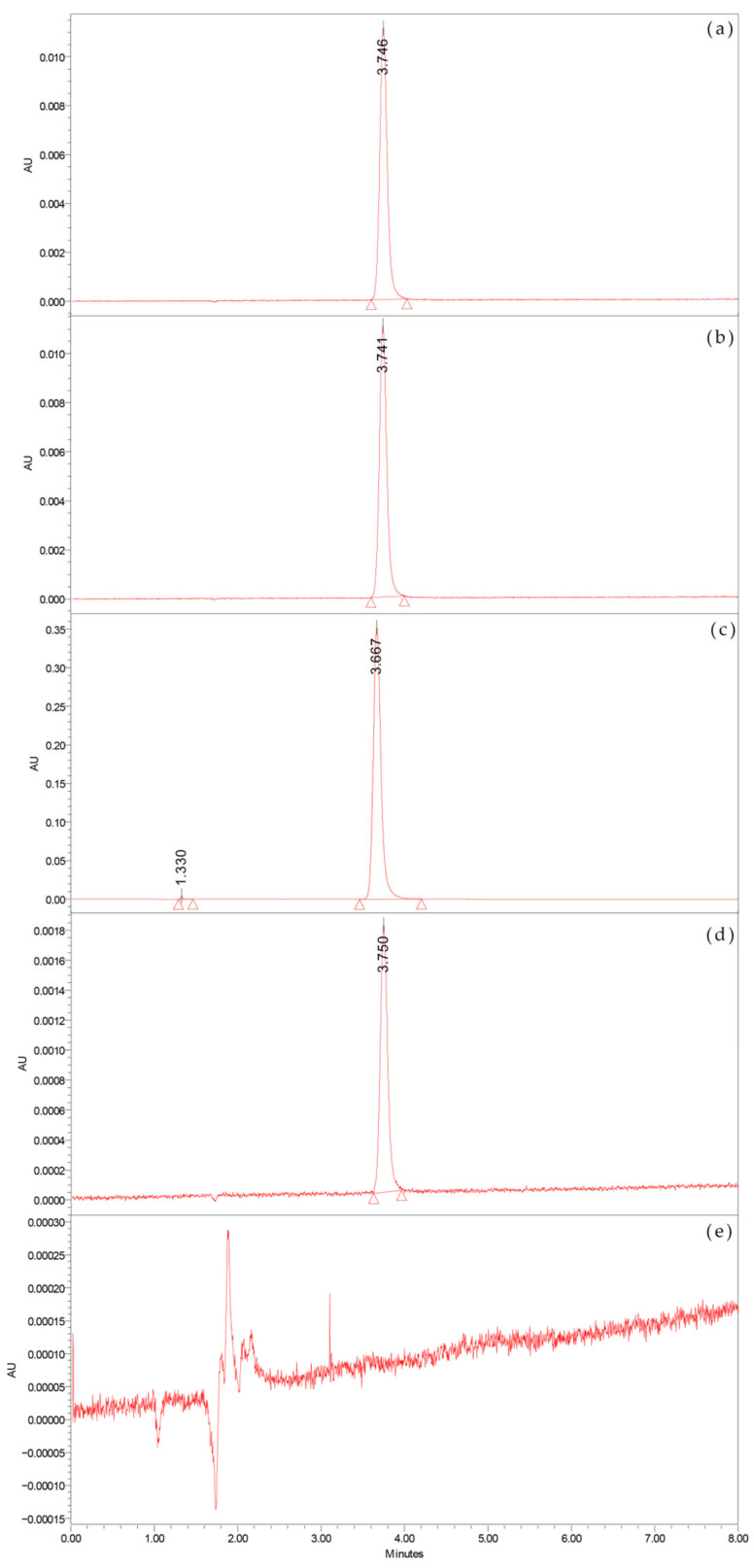
UPLC chromatograms (368 nm) of AA-API (**a**), a photolysis sample (**b**), visible foreign solids (**c**), intermediate polymer, (**d**) and blank excipient (**e**).

**Figure 2 molecules-28-01609-f002:**
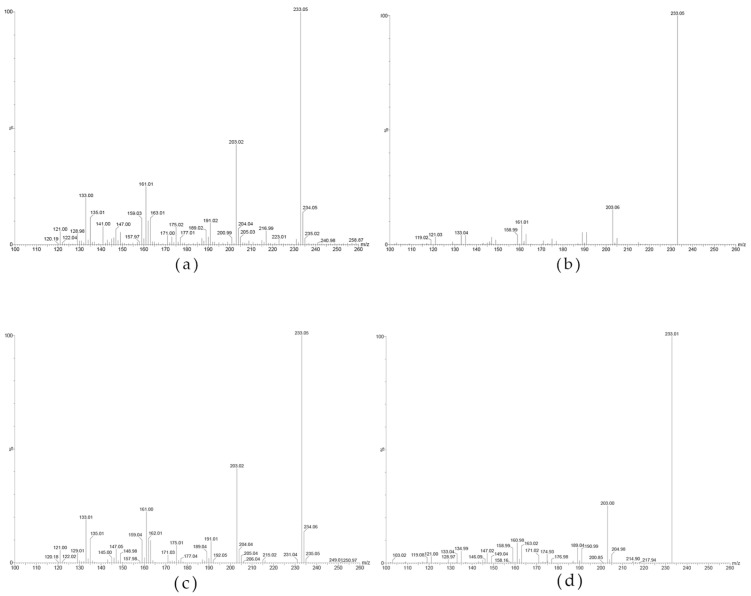
MS spectra (**a**) and MS^2^ spectra (**b**) of AA-API, MS spectra (**c**) and MS^2^ spectra (**d**) of the visible foreign solids (main peaks).

**Figure 3 molecules-28-01609-f003:**
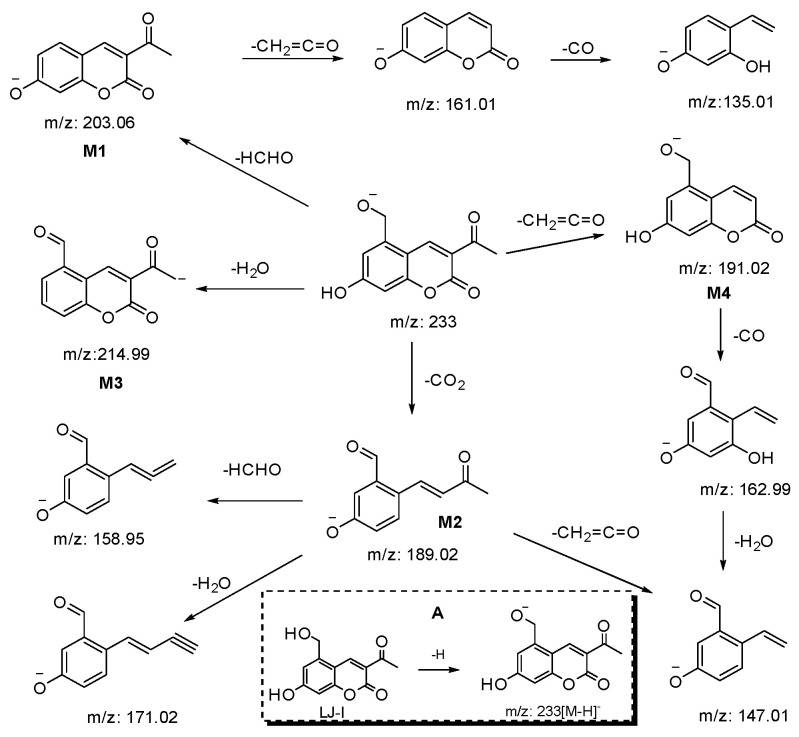
Proposed fragmentation pathways of AA-API and chemical structure of AA-API in negative ion mode as [M-H]^−^.

**Figure 4 molecules-28-01609-f004:**
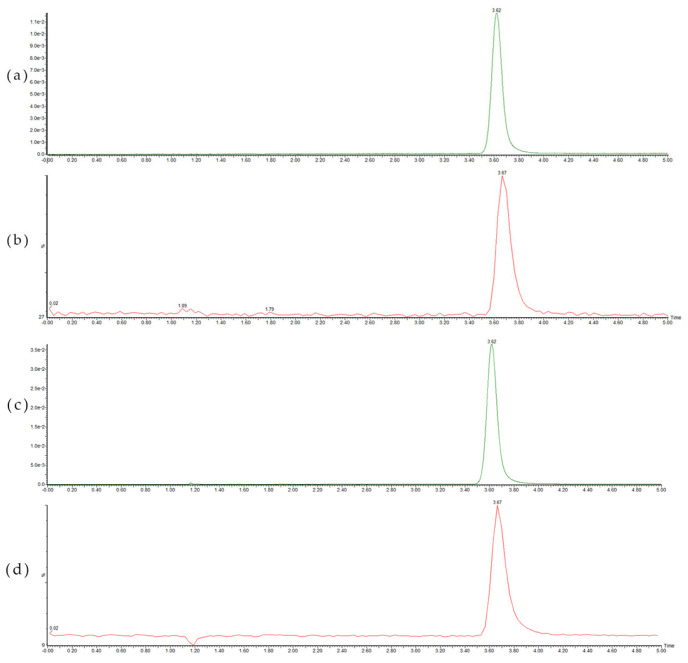
UPLC chromatograms (368 nm) of AA-API (**a**) and the visible foreign solids (**c**), LC-MS total ion current (TIC) of AA-API (**b**) and the visible foreign solids (**d**).

**Figure 5 molecules-28-01609-f005:**
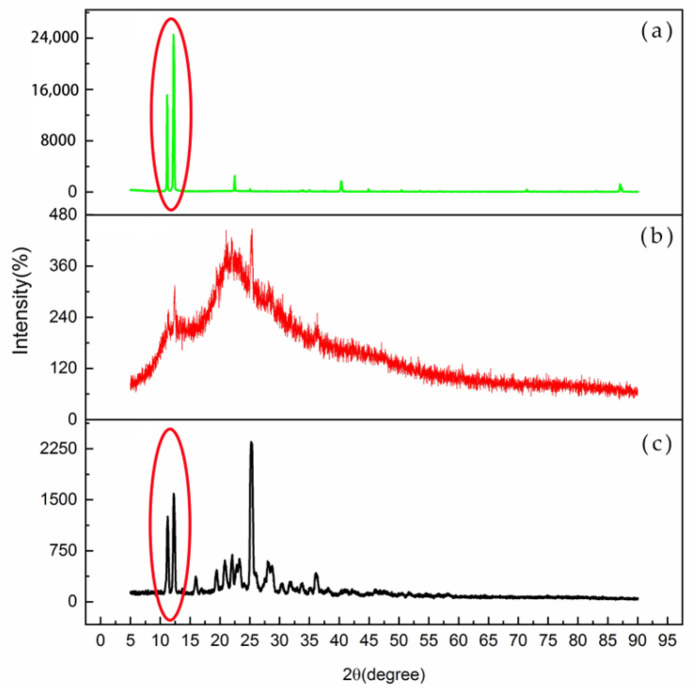
Crystal morphologies of the visible foreign solids (**a**), the intermediate polymer, (**b**) and AA-API (**c**).

## Data Availability

All data is included in the manuscript.

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
