# Peer review of "Qualitative Analysis of Visible Foreign Solids in Armillarisin A Injection Formulations Using Ultra-High Performance Liquid Chromatography–Tandem Mass Spectrometry"

_molecules, 2023, doi:10.3390/molecules28041609_

Round 1

Reviewer 1 Report

The authors have presented a manuscript which is practically describing the challenges the production team faced while producing Armillarisin A injections.

The scientific novelty is not described enough. Is the emphasis on the analytical procedures which were used to identify the impurity, or is it the explanation of the production procedure?

Who is the target audience? Those working in synthesis, production, analysis?

Furthermore, I suggest you change the title of the manuscript. This syntagma "visible foreign solids" is quite misleading. 

The word analysis is also misleading - what you are doing is identifying the drug impurity, which in the end arise from the production procedure.

Here are some further specific comments:

1) why is the abbreviation used for Armillarisin A = LJ?

2) Please add in Figure 1 HPLC-UV chromatogram, with the accompanying wavelength

2) Figure 2 - add a comma after b)

3) Please use the same x-scale in Figure 2 so it is easier to visually analyze spectra

4) where is Figure 6?

5) I am not convinced with the conclusion made in Section 5, line 228-229. How is this proven?

6) Have you tested the conclusion made in the lines 229-232? 

Again, you finish the manuscript with comments regarding the parameters of the production... this goes back to my first question - what is the paper about? production procedure or analytics?

Reviewer 2 Report

Analysis of visible foreign solids in Armillarisin A injection us-2 ing ultra-high performance liquid chromatography-tandem 3 mass spectrometry

Introduction

1.     Please add citations in a sentence in lines 33-35

2.     Please add citations in a sentence in lines 37-43

3.     Please add citation in this sentence “…but occasionally, needle-like yellow-brown crystals are observed 51 during production.” Line 51

4.     Please add an explanation of why needle-like yellow-brown crystals need to be observed and analyzed. Is there any problem or hazard or why?

5.     Please explain more about the urgency and novelty of this research.

Material and Methods

1.     Please add information about where the author got Armillarisin A or  LJ-API, povidone K30, and other reagent or solvent used in this research.

2.     In line 193: “An LJ-I injection was selected ….” What is the mean selected? Selected from or based on what?

3.     “An LJ-I injection was selected and left standing still for 2 min. A syringe was then 193 used to aspirate the upper clear layer, while the large needle-like crystals at the bottom were retained”

I still did not understand or get the point of this research. If the injection of LJ contained “large needle-like crystals at the bottom” can these injection preparations be used?

4.     Are the “large needle-like crystals” like impurities of the compound?

5.     Please explain more about what is the sample. How many samples and is there any preparation for sample or directly measured?

Discussion

1.      Please explain more how is the formulation, how is the production process, and what type and structural characteristic of the API (Armillarisin A) to show and describe the author's explanation that “the visible foreign solids may come from either the API (Armillarisin A) or the polymerization agent (povidone K30)”

2.      “The possible source of the visible solid can be tentatively inferred by combining these results with those already presented in the literature [11–14]”

What does it mean? Please explain more.

Conclusion

1.     “The results revealed that the visible foreign solids was a small amount of LJ-API ingredients not encapsulated by the povidone K30 copolymer”

In the discussion, the author did not explain “not encapsulated”. How an author can conclude this?

Reviewer 3 Report

Armillarisin A is a new coumarin prepared from the mycelium of armillariella tabescens used for for treating acute cholecystitis, chronic cholecystitis, some biliary tract diseases caused and gastritis. Most commercially available injectable formulation with Armillarisin A, a poor soluble compound, uses pharmaceutical-grade propylene glycol as a solvent, but due to its toxicity other types of formulation were needed. Povidone K30, was used as a polymerizing agent in the production of an injectable formulation, but occasionally, needle-like yellow-brown crystals are observed during production. The current study aims to qualitatively identify the visible foreign solids by comparing the UPLC-MS chromatograms and spectra of Armillarisin A. active pharmaceutical ingredient (API), photolyzed samples, intermediate polymer, and excipient and by comparing the X-ray powder diffraction spectra of intermediate polymer and API as well. The results showed that the visible foreign body was attributed to the API itself and that the production process can be improved by optimizing the polymerization process.

The study is well designed and based on the obtained results, the authors can develop further formulation studies on this injectable formulation. The analytical methodology is well described and the results are correct interpreted.  However, some clarifications or adjustments are needed, as follows:

·       It is unclear what does LJ abbreviation stand for.

·       It is unclear if the abbreviation LJ-I is used only for the current povidone formulation or for all injectable formulation containing Armillarisin A.

·       Regarding the preparation of the injectable formulation containing Armillarisin A and Povidone, only a reference to the preparation method of LJ-I National Drug Standard are presented in Materials and Methods. A brief presentation of the preparation method together with the formulation composition should be included in this section and the National Drug Standard should be added to the bibliography section. Also, a reference to the patent mentioned in Section 6 can be made if it is suitable.

·       Line 182: the word “solubility” should be replace with “concentration”.

Reviewer 4 Report

In this paper some information are unclear and some should be added.

In the title of the paper instead of analysis should be qualitative analysis.

L. 16. What is intermediate polymer? Instead of body was detected should be rather body was investigated.

L. 22. What are unpolymerized crystals? Are crystals able to polymerize? 

Fig. 1. What is photolysis sample? What kind of sample it was? What was photolysed? For what it was photolysed? What is intermediate polymer? What is the reason it contain the same substance which is in LJ-API?

Fig. 4. What about the UV detector?

4.2.2. I do not understand the idea of ammonia test. It should be explained. 

Round 2

Reviewer 1 Report

The authors have addressed the comments of the reviewers.

Reviewer 2 Report

I suggest putting your explanation in your "response to reviewer" to the manuscript especially part method and discussion.

The author only answered my question in "response to reviewer" but did not put an explanation in the manuscript.

Thank you.
